# The Impact of Gross Motor Skills on the Development of Emotion Understanding in Children Aged 3–6 Years: The Mediation Role of Executive Functions

**DOI:** 10.3390/ijerph192214807

**Published:** 2022-11-10

**Authors:** Qiaoling Li, Qinglin Wang, Zhaoyang Xin, Huang Gu

**Affiliations:** School of Psychology, Henan University, Kaifeng 475004, China

**Keywords:** gross motor skills, executive functions, emotion understanding, mediating effect, children aged 3–6 years

## Abstract

This study investigates the impact of gross motor skills on the development of emotion understanding and the role of executive function in the relation between gross motor skills and emotion understanding. A total of 662 children were tested for gross motor skills, emotion understanding, and executive function. Regression analysis showed that gross motor skills were significantly related to executive function. Multiple regression analyses showed that gross motor skills and executive function were significant predictors for emotion understanding. Furthermore, mediation analysis showed that executive function mediated the impact of gross motor skills on emotion understanding. Gross motor skills contributed to emotion understanding by improving children’s executive function. The findings imply that a pathway from gross motor skills to emotion understanding is mediated by executive function, which offers a novel perspective on the developmental mechanisms of children’s emotion understanding.

## 1. Introduction

Emotion understanding refers to a conceptual understanding of emotions, including possible causes, subjective feelings, physiological reactions, cognition, resulting action impulses, and a variety of appropriate regulation strategies [1,2,3]. The ability to gain insight and understanding of emotions has important implications for children’s social adaptation, peer acceptance, and prosocial behavior [4,5,6,7,8,9,10]. Previous research has shown that the development of emotion understanding is especially crucial between the ages of 3 and 6, because of the susceptibility of emotion understanding to other abilities during this period [11,12]. However, there is currently limited research on specific factors influencing the ability of emotion understanding, a knowledge gap this study seeks to address by examining the relationship between gross motor skills and executive cognitive functions in children. The relationship between gross motor skills and executive functions in children is the subject of a growing corpus of study [13,14,15,16]. There are also indications that executive function affects emotion understanding [17,18,19]. However, the influence of these variables on emotion understanding has yet to be investigated in children between the ages of 3 to 6. The aim of this study is to reveal the relations between gross motor skills and emotion understanding, and the role played by executive function in this relationship, which will extend our knowledge of a critical aspect of human development. This study provides empirical evidence for a better understanding of the relationship between gross motor skills and executive cognitive function and emotional understanding to make up for the lack of previous research in this area. At the same time, this study proposed the idea of future intervention research for the design of a kindergarten curriculum and activities to provide some reference. In addition, this study also provides a certain reference for parents of young children at home.

The use of large body muscles in balance, limb, and trunk movements is referred to as gross motor skills. Children’s gross motor skills lay the groundwork for the eventual development of more complicated motions and sport-specific talents. Apart from that, gross motor skills are crucial for the development of executive functioning. Children participate in physical activities that demand goal-directed conduct, which helps them develop executive functions [20]. Executive function is an umbrella term for cognitive processes that are involved in purposeful, goal-directed behavior [21]. Three of the core aspects of executive functioning are inhibitory control, working memory and cognitive flexibility [22,23]. The ability to control attention, behavior, and emotions to override a prepotent instinct or a dominant response is known as inhibitory control [21,24]. A working memory, which involves the storage and manipulation of information, and inhibition support each other [25]. The ability to flexibly shift between mental sets is a component of cognitive flexibility, which builds on the first two components—inhibitory control and working memory [26]. These precise executive functions are intertwined, yet they are also obviously distinct [22]. These functions play an important role in children’s development, such as academic achievement [27,28,29].

According to Piaget’s cognitive-developmental theory, sensory and movement activities are crucial for the development of cognitive capacities during the sensorimotor stage [30]. Motor skills provide opportunities for infants to explore and understand the surrounding environment that supports cognitive development. Infants progress in their cognitive development through visual and tactile attention to objects, manipulation of object features, and exploration of the infants’ affects upon the objects [31]. The findings from neurobiological studies also support the link between motor development and executive function. According to research using neuroimaging techniques, during physical and cognitive tasks, brain regions, crucial for motor and cognitive performance, such as the cerebellum, dorsolateral prefrontal cortex, and linking structures (including the basal ganglia), are co-activated. Moreover, motor and cognitive development follow a similar developmental timeline, with both peaking around the preschool years [32,33]. Thus, it is expected that children’s gross motor skills have an impact on the development of executive function. Moreover, previous studies have demonstrated that gross motor skills are significantly related to executive function [13,14,15,16]. Based on previous literature, we hypothesize that children’s gross motor skills will influence the development of executive function.

Previous research has linked the development of emotion understanding to a variety of cognitive capacities, including executive function. According to one study, improvements in several aspects of executive function are crucial for the development of emotion understanding [17]. In several studies, Cohen and colleagues suggested that activation of executive control attenuates emotional interference [34]. Zamora and colleagues also revealed that children between the ages of 8 and 12 reduce neutral and emotional interference in the same way as adults [35]. These studies suggest that emotion is modulated by executive control. According to one study, children with greater inhibitory control were more likely to receive higher ratings on emotion understanding because they need to cultivate an ability to inhibit their own salient but irrelevant action tendency to understand their own and others’ emotions in context [18]. Morra, Parrella, and Camba found that after controlling for age, working memory capacity was an important predictor of the development of emotion understanding [19]. Understanding emotion necessitates that children consider multiple parts of situational, causal, and social knowledge, as well as their understanding of emotion, and to integrate these different types of information to respond appropriately. Working memory is used to integrate all knowledge by storing and manipulating it in the mind. One study also indicated that working memory and inhibition both play a role in false-belief performance, with working memory being a greater predictor of false-belief understanding than inhibition [36]. Cognitive flexibility, on the other hand, permits us to quickly modify our ideas and behaviors in response to a variety of external demands and goals [22,37]. Children must develop capabilities in reflecting on their thoughts and behaviors as well as reacting to their surroundings to grasp their own and others’ viewpoints [38]. Silkenbeumer et al. argue that the development of children’s emotion understanding requires cognitive flexibility to suppress original internal beliefs and to modify external behaviors to adjust to the external environment [39]. Children need to coordinate different information and to flexibly deal with the misleading appeal of certain contextual or emotional cues in order to understand increasingly complicated emotions [19,40]. Furthermore, a growing body of evidence suggests that emotional understanding develops gradually throughout childhood, particularly between the ages of 3 and 6. The age range of 3 to 5 years is particularly crucial for cognitive flexibility development, which is likely to influence the development of emotion understanding. Based on previous research, we contend that the development of emotion understanding is influenced by children’s executive function.

We propose that the pathway from gross motor skills to emotion understanding is mediated by executive function. That is, we contend that gross motor skills may help children understand emotions by enhancing their executive function. To properly explore how gross motor skills may influence emotion understanding, the mediating role of executive function must also be considered, an approach that has been largely overlooked until now.

Based on the extant empirical evidence concerning the relations between gross motor skills, executive function, and emotion understanding, the purpose of this study is to investigate the impact of gross motor skills on the development of emotion understanding and to explore whether executive function mediates the association between gross motor skills and emotion understanding. We presume that there are links between gross motor skills and emotion understanding, although we do not formally propose any clear hypotheses due to a lack of previous research. Furthermore, we expect that executive function mediates the association between gross motor skills and emotion understanding. The findings of this study will be relevant for the screening of children with difficulties in gross motor skills, executive function, and emotion understanding. Furthermore, a greater knowledge of the precise relationships between gross motor skills and emotion understanding will serve as a basis for gross motor skills interventions that may improve aspects of emotion understanding at the same time.

## 2. Materials and Methods

### 2.1. Participants

662 participants (303 females, 359 males) from two preschools in Kaifeng in Henan province, China, took part in this study. Participants were asked to complete a motor skill test and a 30-min E-prime test. After the test, each participant received an age-appropriate present as a token of thanks. Caregivers of every participant signed a written informed consent form after being fully briefed about the nature and purposes of the project. The study protocol was approved by the Psychology Research and Ethics Committee at Henan University in China (ID: 2020226).

### 2.2. Measures

Gross Motor Skills: The Test of Gross Motor Development-Second Edition (TGMD2) was used to test the motor skills of all children aged 3–5 years [41]. TGMD2 consists of two subtests named the locomotor subset and the object control subset. Both subtests consist of six different gross motor skills. For each of these six skills, there are between three and five assessment criteria. The locomotor subset measures running (four criteria), galloping (four criteria), hopping (five criteria), leaping (three criteria), horizontal jumping (four criteria), and sliding (four criteria). The object control subset measures striking a stationary ball (five criteria), stationary dribbling (four criteria), kicking (four criteria), catching (three criteria), overhand throwing (four criteria), and underhand rolling (four criteria). We followed standardized procedures to obtain and compare a child’s score with peers in the normative sample. Gross Motor Quotient (GMO) standard scores were evaluated from the TGMD2 [42,43].

Inhibition Control: Children’s inhibition control was measured using the Children’s Stroop task which consisted of 20 trials [44]. This adaptation requires children to respond to cards with night-time scenes (stars and moon) with the word “day” and to respond to cards with a daytime picture (the sun) with the word “night”. Inhibition was measured based on the number of correct responses given.

Cognitive Flexibility: The Dimensional Change Card Sort task including 36 trials [45] was used to measure cognitive flexibility, which involved target cards that varied along the dimensions of colour and shape (e.g., red and blue, rabbits and boats). After learning to sort the cards according to one dimension (shape or color), participants were asked to sort the cards according to the other dimension. The score represented the number of times the child correctly shifted sets after the sorting criteria changed.

Working Memory: Children’s working memory was measured using Beads tasks. In this task, the 12 cards (red, blue, and white) are presented in a row (There are four shapes: round, triangular, rectangular, oval), and the experiment consists of three blocks, each consisting of five rails. There are two opportunities to practice before the formal experiment. Experiments present the corresponding graphics on the computer screen, and the difficulty increases from one graphic to three graphics. The first block is a graphic, the second block is a combination of two graphics, and the third block is a combination of three graphics. The subjects were asked to indicate the same color and shape as they looked on the screen for a card graphic (or set of graphics) that lasts 2 s (one-bead trials) or 3 s (two-bead trials) or 5 s (three-bead trials). Working memory was measured using the times of correct responses in three parts [46].

Emotion Recognition: Our measure of emotion recognition involved tasks compiled by Bierman and colleagues [47], including emotion matching and emotion identification. Emotion matching involves giving the child four emoticons (happy, angry, sad, and afraid), allowing the child name the target emotion picture (e.g.,: the one test to present a happy picture to the child, and to allow the child to answer this expression); if the child correctly matched the corresponding emotion meter 2 points, the match error meter 0 points. Emotion identification refers to allowing young children to identify the target emotion image from the given emotion picture (e.g.,: the main test simultaneously presents four emotion pictures to the child, respectively, saying “happy, angry, sad, afraid”, so that the child from the four pictures given to point out the corresponding picture), if the child can be identified meter 1 point, identify the error or not to identify meter 0 points. The points gained from these two tests are added to reveal the recognition score.

Emotion Comprehension: The Test of Emotion Comprehension consists of nine components and was developed to measure the comprehension of emotions by children between the ages of 3 and 11. The components were arranged in order of ascending difficulty [12,48]. The narrower age range of participants in this study led us to employ only four of the original nine components of the TEC: (a) external cause, (b) desire, (c) belief, and (d) reminder. The TEC was a picture book with a basic cartoon narrative in which a gender-matched protagonist is confronted with simple or complex situations that trigger various emotional responses. We used the Chinese adaption of the TEC in our study, and it was gender-matched.

While exhibiting a cartoon image of a specific situation, the researcher read a brief statement (e.g., this child has just received a present for his birthday) or a story describing the scenario. The children were then asked to choose one of four drawings of faces depicting various emotional states to reflect how the protagonist feels (happiness, sadness, fear, and anger). The test was divided into four different scenarios, each of which corresponded to a different component and was delivered in a specific order: (a) understanding of external causes of emotions (e.g., attribution of an appropriate emotion to a protagonist receiving a birthday present); (b) understanding of desire-driven emotions (e.g., attribution of an appropriate emotion to two protagonists confronting the same situation but having opposite desires), (c) understanding of emotions based on beliefs (e.g., attribution of an emotion to a boy/girl that is enjoying flowers without realizing that the petals have been destroyed by a dog), and (d) understanding of the influence of a reminder on one’s current emotional state (e.g., attribution of an emotion to a protagonist who is reminded of the flowers destroyed by a dog). In addition, in the last three components of the TEC, some control mechanisms were introduced to help participants to accurately understand the main gist of the scenario. For example, for the scenario involving a birthday party, which corresponds to the comprehension of emotions based on desire, the control question, “Who enjoys the cake?” is designed to elicit the response “The boy enjoys the cake”. If participants answered incorrectly, the researchers provided the correct answers.

### 2.3. Statistical Analysis

Scores from the Inhibition Control, Cognitive Flexibility, and Working Memory tasks were added together to form a new variable: execution function. Before data analysis, all data were standardized. First, using SPSS 21 software, preliminary analyses were conducted to evaluate the descriptive statistics and the association between the study variables. Direct and indirect path parameters and the overall effects were estimated by the bootstrap method, with 5000 resamples to test the mediation model and to calculate the 95% confidence intervals (CIs). When the link between the predictor and mediator variables is significant and the confidence interval does not include zero, the mediator effect is confirmed.

## 3. Results

### 3.1. Descriptive Statistics and Correlations among the Variables

The results of the descriptive statistics and the Pearson correlation coefficient between gross motor skills, cognitive development, emotional comprehension and emotion recognition are listed in Table 1. Regarding the correlations, gross motor skills were significantly correlated with executive function (*r* = 0.37, *p* < 0.01), emotion comprehension (*r* = 0.32, *p* < 0.01), and emotion recognition (*r* = 0.19, *p* < 0.01). In addition, executive function was significantly correlated with emotion comprehension (*r* = 0.36, *p* < 0.010, and emotion recognition (*r* = 0.19, *p* < 0.01). Furthermore, the correlation between emotion comprehension and emotional recognition was significant (*r* = 0.21, *p* < 0.01). The variance inflation factor of the model was smaller than 10, thus indicating no co-linear problem in the sample data.

### 3.2. Test of the Mediating Effect

A regression analysis using the bootstrapping approach by Preacher and Hayes was used to test our propositions, allowing for the simultaneous examination of multiple mediators [49]. Hayes’ PROCESS macro model 4 analysis was performed to investigate the influence of gross motor skills on emotion comprehension and the mediating role of executive function. The results are displayed in Table 2.

The regression model showed that gross motor skills had a significant positive total effect on emotion comprehension (*β* = 0.32, *p* < 0.001). Regarding the mediator, gross motor skills were positively associated with executive function (*β* = 0.37, *p* < 0.001). In addition, executive function had a significant positive direct effect on emotion comprehension (*β* = 0.28, *p* < 0.001). The analysis demonstrated that the direct effect of gross motor skills on emotion comprehension remained significant, but became weaker (*β* = 0.22, *p* < 0.001) when its indirect effects through executive function was taken into account.

The bootstrap estimates for the indirect effect of gross motor skills on emotion comprehension through executive function based on the bootstrap samples of 5000 were also calculated. The confidence interval (CI) was set to 95% to provide conservative results. The mean estimate of the indirect effect through executive function was 0.10 and its 95% CI excluded zero [0.068, 0.143]. The mediating effect accounted for 31.25% of the total effect.

Hayes’ PROCESS macro model 4 analysis was performed to investigate the influence of gross motor skills on emotion recognition and the mediating role of executive function. The results are displayed in Table 2. The regression model showed that gross motor skills had a significant positive total effect on emotion recognition (*β* = 0.19, *p* < 0.001). Regarding the mediator, gross motor skills were positively associated with executive function (*β* = 0.37, *p* < 0.001). In addition, executive function had a significant positive direct effect on emotion recognition (*β* = 0.14, *p* < 0.001). The analysis demonstrated that the direct effect of gross motor skills on emotion recognition remained significant, but became weaker (*β* = 0.14, *p* < 0.001) when its indirect effects through executive function was taken into account.

The bootstrap estimates for the indirect effect of gross motor skills on emotion recognition through executive function based on the bootstrap samples of 5000 were also calculated. The confidence interval (CI) was set to 95% to provide conservative results. The mean estimate of the indirect effect through executive function was 0.05, and its 95% CI excluded zero [0.017, 0.086]. The mediating effect accounted for 26.32% of the total effect.

## 4. Discussion

This study focuses on the role of gross motor skills in the development of emotion understanding. We also wanted to explore whether executive function mediates the relationship between gross motor skills and emotion understanding in children. We found evidence to support previous findings of relations between gross motor skills, executive function, and emotion understanding. First, our regression analysis demonstrated that gross motor skills were a predictor of emotion understanding development. Furthermore, mediation analysis revealed that executive function mediates the relationship between gross motor skills and emotion understanding. That is, early gross motor skills, by improving children’s executive function, contribute to emotion understanding. Our findings offer a new perspective on how children’s understanding of emotions are developed, and they demonstrate the central role of executive function in the development of emotion understanding.

At a behavioral level, the relationships between gross motor skills and executive function can be explained [50,51]. Regular physical activity necessitates and trains gross motor skills and executive processes, as goal-directed conduct is required during various sports and other physical activities. These activities help children adapt to a constantly changing environment. Thus, participation in physical activities not only improves gross motor skills but also helps to enhance executive function [20]. This is compatible with other studies which have found that high competencies in gross motor skills are associated with high performance on specific executive function tasks [14,15,16]. The relationship between gross motor skills and executive function can also be explained through a neuropsychological framework which proposes that there are overlaps in neural networks that are crucial for both gross motor skills and executive function. It is suggested that the cerebellum, dorsolateral prefrontal cortex, and connective structures (including the basal ganglia) all have a role in motor and cognitive activities, making the brain regions co-activated. When one observes decreased dorsolateral prefrontal cortex activation, one also sees a concomitant decrease in cerebellar activation. The activation of these regions is highly linked and intertwined [52,53]. Indeed, the neural pathways that govern gross motor skills and executive function may overlap—particularly in children, where there is a higher degree of brain plasticity, which concluding the caudate pathway of the basal ganglia, sensory-motor cortex etc., particularly when cognitive control of motor activity is considered. These studies support a high correlation based on neurological evidence [32,33].

This study finds that gross motor skills influence emotion understanding by improving children’s executive function. By demonstrating that executive function plays a mediating role in the relationship between gross motor skills and emotion understanding, we advance knowledge in this area, for which previous research had suggested associations between gross motor skills, emotion understanding, and executive function, without pinpointing the precise mechanisms by which they are linked. The development of motor skills in children by scheduling regular physical activities is important in facilitating the development of children’s executive functioning. As other studies have also shown, emotion understanding is associated with performance on specific executive function tasks. For example, Morra and colleagues revealed that a well-developed working memory has a significant impact on the development of one’s emotion understanding [19]. Studies also find that cognitive flexibility is a significant predictor of children’s emotion understanding [54,55]. Furthermore, in emotion comprehension tasks, children need the working memory capacity to memory and integrate various aspects of situational, causal, and social knowledge, as well as their knowledge of emotion. Furthermore, they also need to develop capabilities to reflect on their thoughts and behaviors as well as adjust to the changing environment. In addition, in order to grasp emotions in context, the capacity to block salient but irrelevant action tendencies is also necessary. Executive functioning in children is developed through the training of their gross motor skills, and their ability to understand emotion is also enhanced along the way.

Our findings reveal that gross motor skills predict emotion understanding directly, and also indirectly support its development by improving executive functioning, providing a new perspective on how children’s emotion understanding develops. The strengths of our study include the large sample of children examined, enhancing the generalizability of our results. The three executive function tasks that we examined, and our consideration of executive function as a mediating variable of the relation between gross motor skills and emotion understanding, have helped to refine understandings of this fundamental aspect of child development.

However, there are some limitations to our study. We used a cross-sectional design, making it difficult to assert causal relations between gross motor skills and emotion understanding. Therefore, intervention studies and experimental studies are necessary to investigate the causal relations between gross motor skills and emotion understanding that are inferred in this study. Furthermore, there may be other variables beyond the ones examined in this study which play a role in explaining the relationships we found. Other unmeasured variables, such as the socioeconomic position of the child’s family, could lead to model misspecifications.

## 5. Conclusions

This study found that gross motor skills have an impact on the development of emotion understanding in children and that executive function plays a mediating role in this relationship. We suggest that activities which promote the development of gross motor skills are highly beneficial to children because they also promote the development of executive functioning, which is crucial for emotion understanding. In demonstrating the mediating influence of executive function on the pathway from gross motor skills to emotion understanding, our study offers a novel perspective on the developmental mechanisms of children’s emotion understanding.

## Figures and Tables

**Table 1 ijerph-19-14807-t001:** Descriptive Statistics and Correlations (*n* = 662).

Variables	*M*	*SD*	1	2	3	4
1. Gross Motor Skills	36.63	4.45	1			
2. Executive Functions	60.11	7.16	0.37 **	1		
3. Emotion Comprehension	8.18	1.43	0.32 **	0.36 **	1	
4. Emotion Recognition	16.19	2.85	0.19 **	0.19 **	0.21 **	1

** *p* < 0.01.

**Table 2 ijerph-19-14807-t002:** Results of the regression analysis (*n* = 662).

Variables	Model 1 (EC)		Model 2 (ER)		Model 3 (EF)		Model 4 (EC)		Model 5 (ER)	
	*β*	*SE*	*β*	*SE*	*β*	*SE*	*β*	*SE*	*β*	*SE*
MD	0.32 ***	0.04	0.19 ***	0.04	0.37 ***	0.04	0.22 ***	0.04	0.14 ***	0.04
CD							0.28 ***	0.04	0.14 ***	0.04

*** *p* < 0.001.

## Data Availability

Data available on request due to restrictions privacy. The data presented in this study are available on request from the corresponding author. The data are not publicly available due to privacy.

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
