# Peer review of "The Impact of Gross Motor Skills on the Development of Emotion Understanding in Children Aged 3–6 Years: The Mediation Role of Executive Functions"

_ijerph, 2022, doi:10.3390/ijerph192214807_

Round 1

Reviewer 1 Report

This is a very interesting study.

Well-written and structured with a flow.

It would be nice to add more information on the design. How did the testing take place? where? was it individual participation? what was the sequence of the tests/tasks?

You could also add more on the implications of the study.

Author Response

We truly appreciate your time and effort for reviewing our manuscript. All of your comments have been very helpful to further improve our manuscript. Below, we provide a point-by-point description of all the revisions we made based on the comments and suggestions. 1. It would be nice to add more information on the design. How did the testing take place? where? was it individual participation? what was the sequence of the tests/tasks? Response: After the consultation with the kindergarten, in the cooperation of the kindergarten teachers, we carried out the test of this study in the kindergarten. For testing convenience, the gross motor skills test was conducted separately from the executive function test and the emotion test. Every child participated in the test individually. As for the sequence of the tests, the executive function and emotion were tested after the children took part in the test of gross motor skills. Consider the difficulty of working memory, the child should participated in the test of working memory firstly. Then, the child would took part in the test of inhibition control, cognitive flexibility, emotion recognition, and emotion comprehension. 2. You could also add more on the implications of the study. Response: Thank you for your comments. We have revised the paper according to your comments in the manuscript.

Reviewer 2 Report

The current paper “The impact of gross motor skills on the development of emotion understanding in children aged 3-6 years: The mediation role of executive function” provides supportive findings that indicate an important association between gross motor skills and identifying emotions, possibly affected by executive function in children between 3-6 years. The study provides significant data that may contribute to designing and informing the use of gross motor skill developmental programs that may also improve emotional understanding in children.  

However, there are some concerns and suggestions.

As the paper currently stands, it is a cross-sectional study, therefore not mechanistic inferences or cause-and-effect conclusions can be drawn. Granted, the statistical approach is well chosen, and may show independent associations and contributions, there are several aspects that should be considered in analyses. Age-adjusted and sex adjusted statistical analyses (in cases where the testing material was not gender matched)  as well as the basic characteristics of the study population in terms of the actual scores and results of each cognitive and motor tests, should be considered. Additionally, have the effects of other mediators of cognitive function in children been taken into account, such as nutrition, parent-child interaction, socio-demographic, sleep duration differences etc.?

Additional concerns and suggestions:

Introduction:

The introduction is well formulated, and the authors have done a great job in identifying and highlighting the need for the current research project.

However, some of the interpretations and discussions in the introduction, might be best suited in the Discussion section to support of the authors’ data. All in all, the literature is sufficient, but certain aspects will be more appropriate in the Discussion.

Minor aspects:

Page 1, line 25: “cognitions” should possibly “cognition”

Page 1, line 33: executive cognitive functions, rather than only cognitive function

Methods:

It is clear that the authors considered major confounders, excellent skill and cognitive test which were age and gender (in some cases) appropriate. In addition the statistical approach is well chosen, and speaks to a thorough study design consideration. Of significant concern is the lack of mention of ascent from the children included (although they cannot give consent, a form of ascent should be provided and indicated by the children in an age-applicable manner). Age-adjusted and sex adjusted statistical analyses (in cases where the testing material was not gender matched)  - particularly age-adjusted, as cognitive executive function and gross motor skills may differ significantly in a child of 3 versus 6 years.

 Discussion:

 The authors have formulated a great discussion, with literature that supports their findings.

Minor changes and suggestions include:

 Focusing on interpreting their own results within the context of current literature. It is also suggested to rather use the term “comparable” to “compatible” when referring to literature that supports current findings.

 Suggest that the descriptive language is revised, to ensure the clear and effective communication of these interesting results reported by the authors.

Page 6, line 294: Indeed the neural pathways that govern gross motor skills and executive function may overlap – particularly in children, where there is a higher degree of brain plasticity. Particularly when cognitive control of motor activity is considered, the caudate pathway of the basal ganglia, sensory-motor cortex etc.

Page 7, line 302: As this is a cross-sectional observation, one cannot infer that there is improvement of executive function by any means. A relationship exists, indeed, however, whether it be causal in nature, cannot be stated using this study design.

Page 7, line 314: A very important aspect to consider, particularly in children, is the role of the amygdala in the emotional recognition and executive function.  Some studies have shown specific volume difference regarding the hippocampus, orbitofrontal cortex, and particularly the amygdala. Tis is very true in children that have experienced early life trauma or stressful events. Have the authors considered the possible impact of early life trauma/ stress on the results?

 Minor aspect:

The manuscript should please be revised by a language editing service to ensure optimal communications and descriptions.

Author Response

We truly appreciate your time and effort for reviewing our manuscript. All of your comments have been very helpful to further improve our manuscript. Below, we provide a point-by-point description of all the revisions we made based on the comments and suggestions. 1. As the paper currently stands, it is a cross-sectional study, therefore not mechanistic inferences or cause-and-effect conclusions can be drawn. Granted, the statistical approach is well chosen, and may show independent associations and contributions, there are several aspects that should be considered in analyses. Age-adjusted and sex adjusted statistical analyses (in cases where the testing material was not gender matched) as well as the basic characteristics of the study population in terms of the actual scores and results of each cognitive and motor tests, should be considered. Additionally, have the effects of other mediators of cognitive function in children been taken into account, such as nutrition, parent-child interaction, socio-demographic, sleep duration differences etc.? Response: In this study, we used a cross-sectional design, making it difficult to assert causal relations between gross motor skills and emotion understanding. We have discussed this in the manuscript, we also put forward some improvements that need to be made in future studies. We put forward that intervention studies and experimental studies were necessary to investigate the causal relations between gross motor skills and emotion understanding that were inferred in this study. As for the statistical approach, before the data analysis, all data were standardized, we used the standardized scores for data analysis. Although we did not consider the effect of sex and age on this study when analyzing the data, the focus of this study was to explore the relationship between the variables, with reference to previous relevant literature, there was no control for the effects of sex and age, so sex and age were not considered as control variables in the data analysis. As for the effects of other mediators of cognitive function in children, in this study, we didn’t test these variables, such as nutrition, parent-child interaction, socio-demographic, sleep duration differences etc., so we couldn’t control the effects of these variables on this study. This is also a weakness of this study, which needs to further consider the impact of these variables in future studies, and to treat these variables as control variables when data are analyzed. 2. The introduction is well formulated, and the authors have done a great job in identifying and highlighting the need for the current research project. However, some of the interpretations and discussions in the introduction, might be best suited in the Discussion section to support of the authors’ data. All in all, the literature is sufficient, but certain aspects will be more appropriate in the Discussion. Response: Thank you for your comments. We have revised the paper according to your comments in the manuscript. 3. Minor aspects: Page 1, line 25: “cognitions” should possibly “cognition” Page 1, line 33: executive cognitive functions, rather than only cognitive function Response: Thank you for your comments. We have revised the paper according to your comments in the manuscript. 4 Methods: It is clear that the authors considered major confounders, excellent skill and cognitive test which were age and gender (in some cases) appropriate. In addition the statistical approach is well chosen, and speaks to a thorough study design consideration. Of significant concern is the lack of mention of ascent from the children included (although they cannot give consent, a form of ascent should be provided and indicated by the children in an age-applicable manner). Age-adjusted and sex adjusted statistical analyses (in cases where the testing material was not gender matched) - particularly age-adjusted, as cognitive executive function and gross motor skills may differ significantly in a child of 3 versus 6 years. Response: As for the statistical approach, before the data analysis, all data were standardized, we used the standardized scores for data analysis. Although we did not consider the effect of sex and age on this study when analyzing the data, the focus of this study was to explore the relationship between the variables, with reference to previous relevant literature, there was no control for the effects of sex and age, so sex and age were not considered as control variables in the data analysis. This is also a weakness of this study, which needs to further consider the impact of these variables in future studies, and to treat these variables as control variables when data are analyzed. 5. Focusing on interpreting their own results within the context of current literature. It is also suggested to rather use the term “comparable” to “compatible” when referring to literature that supports current findings. Suggest that the descriptive language is revised, to ensure the clear and effective communication of these interesting results reported by the authors. Response: Thank you for your comments. We have revised the paper according to your comments in the manuscript. 6. Page 6, line 294: Indeed the neural pathways that govern gross motor skills and executive function may overlap – particularly in children, where there is a higher degree of brain plasticity. Particularly when cognitive control of motor activity is considered, the caudate pathway of the basal ganglia, sensory-motor cortex etc. Response: Thank you for your comments. We have revised the paper according to your comments in the manuscript. 7. Page 7, line 302: As this is a cross-sectional observation, one cannot infer that there is improvement of executive function by any means. A relationship exists, indeed, however, whether it be causal in nature, cannot be stated using this study design. Response: In this study, we used a cross-sectional design, making it difficult to assert causal relations between gross motor skills and emotion understanding. We have discussed this in the manuscript, we also put forward some improvements that need to be made in future studies. In fact, in this study, it is more to explore the relationship between variables. We put forward that intervention studies and experimental studies are necessary to investigate the causal relations between gross motor skills and emotion understanding in the future. 8. Page 7, line 314: A very important aspect to consider, particularly in children, is the role of the amygdala in the emotional recognition and executive function. Some studies have shown specific volume difference regarding the hippocampus, orbitofrontal cortex, and particularly the amygdala. This is very true in children that have experienced early life trauma or stressful events. Have the authors considered the possible impact of early life trauma/ stress on the results? Response: In this study, we analyzed the data with the standard scores of behavioral indicators of executive function and emotional recognition. We did not consider the brain mechanisms of executive function and emotion recognition in this study. So we didn’t show the specific volume difference regarding the hippocampus, orbitofrontal cortex, and particularly the amygdala. As for the early life trauma or stressful events, we all choose the children that have not experienced early life trauma or stressful events as our participant. Therefore, we didn’t consider the possible impact of early life trauma/ stress on the results. This is also a weakness of this study, which needs to further consider the impact of early life trauma/ stress on the results in future studies, and to treat these variables as control variables when data are analyzed. 9. The manuscript should please be revised by a language editing service to ensure optimal communications and descriptions. Response: Thank you for your comments. We have asked native English speakers to polish the manuscript.
